# ARE ALL NEGATIVES CREATED EQUAL IN CONTRASTIVE INSTANCE DISCRIMINATION?

## ABSTRACT

Self-supervised learning has recently begun to rival supervised learning on computer vision tasks. Many of the recent approaches have been based on contrastive instance discrimination (CID), in which the network is trained to recognize two augmented versions of the same instance (a *query* and *positive*) while discriminating against a pool of other instances (*negatives*). The learned representation is then used on downstream tasks such as image classification. Using methodology from MoCo v2 (Chen et al., 2020c), we divided negatives by their difficulty for a given query and studied which difficulty ranges were most important for learning useful representations. We found a minority of negatives—the hardest 5%—were both necessary and sufficient for the downstream task to reach nearly full accuracy. Conversely, the easiest 95% of negatives were unnecessary and insufficient. Moreover, the very hardest 0.1% of negatives were unnecessary and sometimes detrimental. Finally, we studied the properties of negatives that affect their hardness, and found that hard negatives were more semantically similar to the query, and that some negatives were more consistently easy or hard than we would expect by chance. Together, our results indicate that negatives vary in importance and that CID may benefit from more intelligent negative treatment.

## 1 INTRODUCTION

In recent years, there has been tremendous progress on *self-supervised learning* (SSL), a paradigm in which representations are learned using a *pre-training task* that uses only unlabeled data. These representations are then used on *downstream tasks*, such as classification or object detection. Since SSL pre-training does not require labels, it can leverage unlabeled data, which is can be abundant and cheaper to obtain than labeled data. In computer vision, representations learned from unlabeled data have historically underperformed representations learned directly from labeled data. Recently, however, newly proposed SSL methods such as MoCo (He et al., 2019; Chen et al., 2020c), SimCLR (Chen et al., 2020a;b), SwAV (Caron et al., 2020), and BYOL (Grill et al., 2020) have dramatically reduced this performance gap.

The MoCo and SimCLR pre-training tasks use *contrastive instance discrimination* (CID), in which a network is trained to recognize different augmented views of the same image (sometimes called the *query* and the *positive* and discriminate between the query and the augmented views of other random images from the dataset (called *negatives*).[1])

Despite the empirical successes of CID, the mechanisms underlying its strong performance remain unclear. Recent theoretical and empirical works have investigated the role of mutual information between augmentations (Tian et al., 2020), analyzed properties of the learned representations such as alignment and uniformity (Wang & Isola, 2020), and proposed a theoretical framework (Arora et al., 2019), among others. However, existing works on CID have not investigated the relative importance or semantic properties of different negatives, even though negatives play a central role in CID. In other areas, works on hard negative mining in metric learning (Kaya & Bilge, 2019) and on the impact of different training examples in supervised learning (Birodkar et al., 2019) suggest that understanding the relative importance of different training data can be fruitful.

---

[1]In MoCo, these are called the *query* and *positive* and are treated slightly differently; in SimCLR, both are treated the same and are called *positives*. The other SSL methods listed (not SimCLR and MoCo) are not CID.

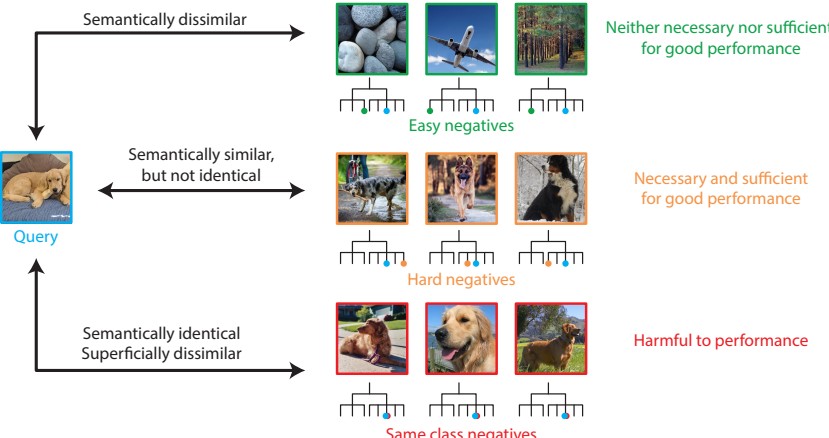

**Figure 1: Schematic summary of main results.** Easy negatives are unnecessary and insufficient (green) and are more often dissimilar (i.e., in unrelated ImageNet classes) to the query (light blue) compared to harder negatives. Hard (but not the very hardest) negatives were necessary and sufficient (orange) and are more often semantically similar to the query compared to easier negatives. The very hardest negatives are unnecessary and sometimes detrimental and also are more often in the same class as the query, compared to easier negatives (red). This is an illustrative schematic; images and trees are not from ImageNet.

In this work, we empirically investigate how the *difficulty* of negatives affects the downstream performance of the learned representation. We measure difficulty using the dot product between the normalized contrastive-space embeddings of the query and the negative. A dot product closer to 1 suggests a negative that is more difficult to distinguish from the query. We ask how different negatives, by difficulty, affect training. Are some negatives more important than others for downstream accuracy? If so, we ask: Which ones? To what extent? And what makes them different?

We focus on MoCo v2 (Chen et al., 2020c) and the downstream task of linear classification on ImageNet (Deng et al., 2009), and have similar results for SimCLR in Appendix A.2. We make the following contributions (see Figure 1 for summary):

- **The easiest 95% of negatives are unnecessary and insufficient, while the top 5% hardest negatives are necessary and sufficient**: We reached within 0.7 percentage points of full accuracy by training on the 5% of hardest negatives for each query, suggesting that the 95% easiest negatives are unnecessary. In contrast, the easiest negatives are insufficient (and, therefore, the hardest negatives are necessary): accuracy drops substantially when training on only the easiest 95% of negatives. The hardest 5% of negatives are especially important: training on only the next hardest 5% lowers accuracy by 15 percentage points.

- **The hardest 0.1% of negatives are unnecessary and sometimes detrimental**: Downstream accuracy was unchanged or improved when we removed these hardest negatives. These negatives were more often in the same ImageNet class as the query, compared to easier negatives, suggesting that semantically identical (but superficially dissimilar) negatives were unhelpful or detrimental.

- **Properties of negatives**: Based on our observations that the importance of a negative varies with its difficulty, we investigate the properties of negatives that affect their difficulty.

  - We found that hard negatives were more semantically similar to the query than were easy negatives: the hardest 5% of negatives were more likely to be of the same ImageNet class as the query, compared to easier negatives. These hard negatives were also closer to the query as measured by depth of the least common ancestor for the negative and the query in the WordNet tree (which ImageNet is built upon).

  - We also observed that the pattern is reversed for the ≈50% of easier negatives: there, the easier the negative, the more semantically similar it is to the query.

  - There exist negatives that are more consistently hard across queries than would be expected by random chance.

We emphasize that our primary aim is to better understand the differences between negatives and the impact of these differences on existing methods rather than to propose a new method. However, our results suggest that there may be unexploited opportunities to reduce the cost of modern CID methods (Chen et al., 2020c). For any particular query, only a small fraction of the negatives are necessary. Our observations on differences in negative importance can serve as a valuable building block for the understanding and improvement of contrastive learning methods.For example, there may be further room to choose specific examples for training, similar to hard negative mining and curriculum learning (Chen et al., 2020a; Chuang et al., 2020; Kaya & Bilge, 2019), to reduce costs and improve performance per data sample.

## 2   METHODS AND PRELIMINARIES

**Contrastive instance discrimination and momentum contrast.** Momentum Contrast (MoCo v2) is a CID method that reaches accuracy within 6 percentage points of supervised accuracy on ImageNet with ResNet-50 (Chen et al., 2020c). In MoCo, the task is to learn a representation that succeeds at the following: given a *query* (an augmented view of an image), correctly pick a *positive* (a different augmented view of the same image) from a large set of *negatives* (augmented views of randomly chosen images). Our experiments focus on aspects that are common between CID methods rather than those specific to MoCo. We discuss implementation details that may be specific to MoCo v2 here.

The MoCo v2 encoder is a ResNet-50 network. For pre-training, the outputs of this base network are fed into a multi-layer perceptron (MLP) head; we refer to the normalized output from the MLP head as the *contrastive-space embedding*. For downstream tasks, the MLP head is discarded and only the base network is used; we refer to the output of the base network as the *learned representation*. A distinguishing feature of MoCo is that it has two encoders, one of which is actively trained (used for the query) and the other which is a moving average of the trained encoder (used for the positive and negatives). MoCo stores the embeddings of each batch of positives in a large queue and uses them as negatives for future batches, enabling the use of more negatives than can fit in a batch.

MoCo uses the InfoNCE loss (Gutmann & Hyvärinen, 2010; van den Oord et al., 2018):

$$\mathcal{L}_q = -\log \frac{\exp(q \cdot k_+/\tau)}{\sum_{i=1}^{K} \exp(q \cdot k_i/\tau)}$$

where $q$ is the embedding of a query (using the learned encoder), $k_+$ is the embedding of a positive (using the momentum encoder), and $k_i$ are the embeddings of the negatives in the queue (added using previous states of the momentum encoder). $\tau$ is a temperature hyperparameter.

**Difficulty of negatives.** To compute the difficulty for a set of negatives given a particular query, we calculate the dot product between the normalized contrastive-space embedding of each negative with the normalized contrastive-space embedding of the query. We then sort the dot products and consider the negatives with dot products closer to 1 to be *harder negatives* and those with smaller dot products to be *easier negatives*. We use this terminology because it fits intuition: all else being equal, harder negatives increase the loss. Since embeddings are normalized, the dot product is the cosine of the angle between the embeddings of the instances and ranges from -1 to 1.

Note that difficulty is defined *per query* and that it is a function of the current state of the network. Thus, a negative can be easy for some queries and hard for others, and the hardness of a negative for a given query can vary over training epochs and across different training runs and configurations.

**Experimental setting.** Our experiments focus on MoCo v2 (Chen et al., 2020c), an improved version of MoCo which combines MoCo v1 (He et al., 2019) with several features of SimCLR (Chen et al., 2020a). We use ImageNet for pre-training and evaluate performance using linear classification on ImageNet from the representation learned in the pre-training CID task. The network used, as in MoCo v2, is a ResNet-50 with MLP head, and trained for 200 epochs. Unless otherwise noted, we run each experiment three times with different seeds; error bars represent mean $\pm$ standard deviation.

**Figure 2: Easy negatives are neither necessary nor sufficient, while hard negatives are both necessary and sufficient.** **a-b**) Top-1 (**a**) and Top-5 (**b**) performance of networks trained on *only* segments of 5% of negatives ordered by difficulty. For example, 95-100% means that only the top 5% hard negatives were used for training. **c-d**) Top-1 (**c**) and top-5 (**d**) performance of networks trained on increasingly larger fractions of the easiest negatives. Error bars are standard deviation across 3 seeds.

# 3  WHICH NEGATIVES ARE NECESSARY OR SUFFICIENT?

We examine which negatives, by difficulty, are necessary or sufficient for learning representations that have strong downstream performance. Outside of CID, there are varying perspectives on the value of easy negatives. Research on hard negative mining suggests that harder negatives can be more important than easier negatives for relevant tasks (Kaya & Bilge, 2019). However, in some supervised contexts, much or all training data seems important for reaching the highest accuracy (Birodkar et al., 2019). We aim to experimentally assess which of these perspectives applies when using MoCo v2 on ImageNet classification.

To determine whether a set of negatives was necessary, we removed the corresponding negatives on each pre-training step; if the resulting representations still led to accuracy close to baseline on the downstream task, then we considered those negatives to have been unnecessary. To determine whether a set of negatives was sufficient, we removed all negatives *except* those in that range on each pre-training step; if the resulting representations still led to strong accuracy on the downstream task, then we considered the negatives in that range to have been sufficient.[2]

**The easy negatives are unnecessary; the hard negatives are sufficient.** First, we asked whether the easy negatives were necessary (or equivalently, whether the hard negatives were sufficient). That is, does the network maintain downstream accuracy when it is pre-trained without the easy negatives? To test this, we evaluated how accuracy changed as different subsets of negatives were removed. Interestingly, we found that using only the hardest 5% of negatives was largely sufficient to recover baseline accuracy (Figure 2a-b, 95-100%), suggesting that the overwhelming majority of the easier negatives were unnecessary. Moreover, the hardest 5% (95-100%) were substantially more informative than the next 5% (90-95%): top-1 accuracy dropped by only 0.7 percentage points when trained on only the hardest 5% vs. 15 percentage points for the next hardest 5% (90-95%) and 47 percentage points for the third 5% (85-90%; Figure 2a-b). Going forward, we use 5% as a cutoff and call the negatives harder than this cutoff *hard* and those easier than this cutoff *easy*.

**The easy negatives are largely insufficient; the hard negatives are necessary.** We next asked whether the easy negatives were sufficient (or, equivalently, whether the hard negatives were necessary). Although we found in the previous section that the easy negatives were unnecessary, that does not necessarily mean they are insufficient. For example, it could be that the easy negatives, while individually less important, collectively provide sufficient signal for learning good representations on the downstream task. Alternatively, it is possible that the information contained in all of the easy negatives still lacks enough signal; in this case, the easy negatives, even when combined together, would still be insufficient.

We found that even when the easiest 95% of negatives were combined together, accuracy was 5.4% below baseline (Figure 2c-d). In contrast, recall that using only the hardest 5% of negatives (19x fewer) achieved top-1 performance within 0.7% of baseline (Figure 2a). Using the easiest 90% of negatives harms accuracy even further (0-90%; Figure 2c-d). Together, these results demonstrate that the easiest negatives, even when they comprise the vast majority of negatives, are still insufficient.

**The very hardest negatives are harmful at lower temperatures.** We have found that the *hard* negatives, i.e. the 5% hardest, are largely necessary and sufficient for CID. However, top-1 accuracy actually *improved* slightly when we removed the very hardest 0.1% of negatives ($p = 0.03$ for an

---

[2]We removed sets of negatives by treating them as through they were not present in the queue.

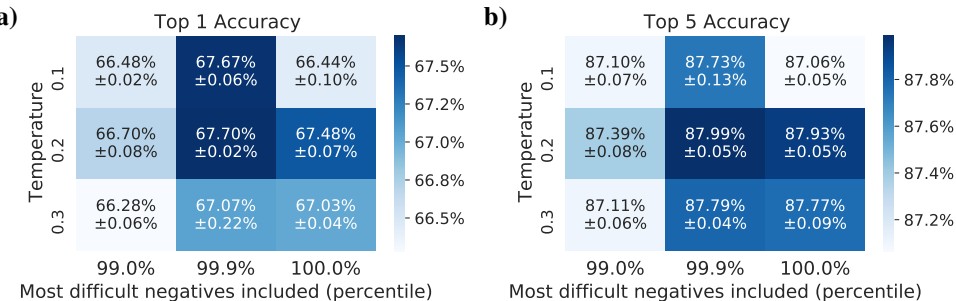

**Figure 3: The hardest 0.1% of negatives hurt, especially at lower temperatures.** Top-1 (**a**) and top-5 (**b**) accuracy of networks trained on all but hard and hardest negatives, at different temperatures. The baseline temperature is 0.2.

unpaired t-test).[3] This effect was most pronounced at lower temperatures (Figure 3); for example, at temperature 0.1 (as opposed to baseline temperature 0.2), training without the hardest 0.1% of negatives improved downstream top-1 accuracy by 0.23% ($p = 0.0003$) and top-5 accuracy by 0.67%. One might expect such a difference between temperatures because the hardest negatives are weighted more in the loss at lower temperatures. Interestingly, the effect was larger for top-5 accuracy than top-1 accuracy (compare Figure 3b with 3a).

One hypothesis for why the hardest negatives hurt is that because negatives are randomly sampled, some negatives can be too similar to the query (e.g. augmentations of near-duplicates to the query). Because negatives are randomly sampled, they can included augmented views of images that are near-duplicates of the query or otherwise visually very similar to the query. If both the query and negative contain identical semantic content, the contrastive loss might rely on non-semantic features to distinguish between them, thus emphasizing these non-semantic features in the representation (Figure 1). These same-class negatives may thus be harmful to learning representations for downstream linear classification.

If this is the case, we would expect that removing same class negatives would improve performance, perhaps even more than removing the hardest 0.1% of negatives overall. As shown in Table 1, removing same-class negatives indeed leads to slightly higher accuracy than removing the hardest 0.1% of negatives. Removing only the subset of the hardest 0.1% of negatives with the same class as the query accounts for all of the improvement from removing the hardest 0.1% of negatives. Alternatively, removing only the subset of the 0.1% hardest negatives with *different* classes shows no improvement over baseline and in fact decreases top-1 accuracy at low temperature.

These results demonstrate that the accuracy benefit of removing the 0.1% hardest negatives can entirely be accounted for by the fact that it removes many elements of the same class as the query, so that removing the 0.1% hardest negatives approximates removing the same-class negatives without requiring access to privileged label data. This observation is also consistent with recent work which has attempted to "debias" contrastive learning away from same-class negatives (Chuang et al., 2020).

## 4 UNDERSTANDING NEGATIVES BY DIFFICULTY

**Hard negatives are more semantically similar to the query.** We have shown that easy negatives are unnecessary and insufficient, and that, inversely, hard negatives are necessary and sufficient. However, the properties that distinguish easy from hard negatives are not yet understood. Intuitively, we might imagine that to learn a representation that is useful for a fine-grained classification task such as ImageNet, a network must learn to distinguish between categories that are similar but semantically distinct, e.g., different breeds of dogs. If this were the case, we would expect that the 5% hardest negatives, which were both necessary and sufficient for training, would also be more semantically similar to the query than the 95% easiest negatives.

---

[3]For this section, to remove a set of negatives, we replace them with slightly older negatives, so that the total number of negatives used does not change. To accommodate this change, the queue is made slightly larger, with the additional length remaining unused except to replace negatives we want to remove.

|  | Temperature = 0.07 | | Temperature = 0.2 | |
|---|---|---|---|---|
|  | Top-1 Acc | Top-5 Acc | Top-1 Acc | Top-5 Acc |
| Baseline (remove none) | $64.78 \pm 0.31$ | $85.86 \pm 0.12$ | $67.48 \pm 0.07$ | $87.93 \pm 0.05$ |
| Remove 0.1% hardest | $66.25 \pm 0.23$ | $86.98 \pm 0.09$ | $67.64 \pm 0.22$ | $87.88 \pm 0.07$ |
| Remove same class | $66.61 \pm 0.10$ | $86.96 \pm 0.07$ | $68.07 \pm 0.12$ | $88.30 \pm 0.15$ |
| Remove 0.1% hardest $\cap$ same class | $66.43 \pm 0.04$ | $86.78 \pm 0.06$ | $67.67 \pm 0.02$ | $88.09 \pm 0.18$ |
| Remove 0.1% hardest $\cap$ different class | $63.69 \pm 0.04$ | $85.44 \pm 0.00$ | $67.38 \pm 0.06$ | $87.86 \pm 0.08$ |
| Remove 99.9% easiest $\cap$ same class | $65.06 \pm 0.11$ | $85.91 \pm 0.01$ | $67.79 \pm 0.07$ | $88.05 \pm 0.05$ |

Table 1: **The hardest 0.1% negatives hurt because of same-class negatives**: Downstream accuracy when removing negatives of same/different class as the query and easier/hardest negatives at different temperatures. At temperature 0.07, accuracy improves when removing same-class negatives and/or hard negatives. At temperature 0.2 (default), there is a similar but smaller effect.

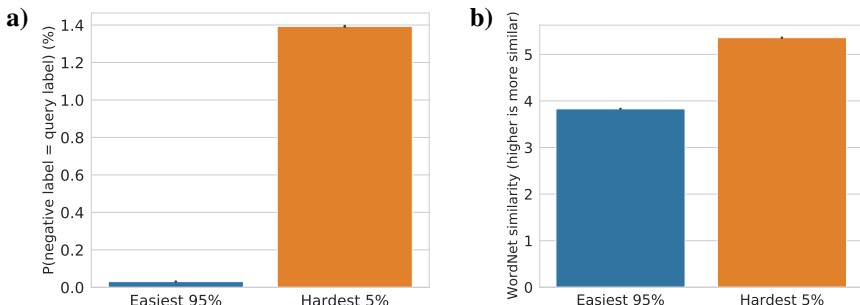

Figure 4: **Semantic similarity is higher for the 5% of hard negatives than for the 95% of easy negatives** Proportion of shared labels (higher is more similar) (**a**) and WordNet distance as depth of least common ancestor (lower is more similar) (**b**) for the 5% of hard negatives and the 95% of easy negatives. Error bars are standard deviation on plot values across 3 seeds.

To test this hypothesis, we first examined the fraction of the easy and hard negatives that had the same class as the query label.[4] Similar to our results above regarding the 0.1% very hardest negatives, we found that negatives of the same class were significantly overrepresented among the 5% hardest negatives relative to the easy negatives ($p = 5.1 \cdot 10^{-7}$, unpaired t-test; Figure 4a).

However, this experiment can only tell us whether the hard negatives contain more negatives that are *semantically identical* to the query (in that they have the same class); it cannot distinguish between negatives of different semantic similarity with the query (which have classes that are related, but distinct from the query). To evaluate semantic similarity we used the ImageNet class hierarchy derived from WordNet (Deng et al., 2009). For each negative, we computed the tree depth of the least common ancestor between the negative and the query; higher WordNet similarity means that the least common ancestor is deeper in the tree and that the negative is therefore more similar to the query.[5] As shown in Figure 4b, we found that the hard negatives were significantly more semantically similar to the query than the easy negatives (p=4.8e-7, unpaired t-test). Together, these results demonstrate that semantic similarity is a property that distinguishes easy and hard negatives; however, evaluation of whether this relationship is causal is left for future work.

**Some of the easiest negatives are both anti-correlated and semantically similar to the query.** Perhaps surprisingly, we also found that a small subset of the very easiest examples are *anti-correlated* with the query (i.e., the dot product between these negatives and the query is highly negative; Figure 5c). While the presence of negatives orthogonal to the query might be expected (as the two might be unrelated to one another) the presence of a high magnitude negative dot product suggests that the network learned to anti-correlate these negatives with the query.

Moreover, these negatives are also substantially more semantically similar to the query than the majority of easy negatives (Figure 5b); in fact, by the WordNet tree similarity (depth of least common ancestor), their semantic similarity nearly matches those of the hard negatives. In addition, qualitatively, the positive and negative classes with the highest mean pairwise negative dot product are

---

[4]In this section, use 2K query images and 2K negatives; use trained non-momentum encoder at 200 epochs.
[5]Although WordNet similarity is not a perfect measure, we do believe it is valuable.

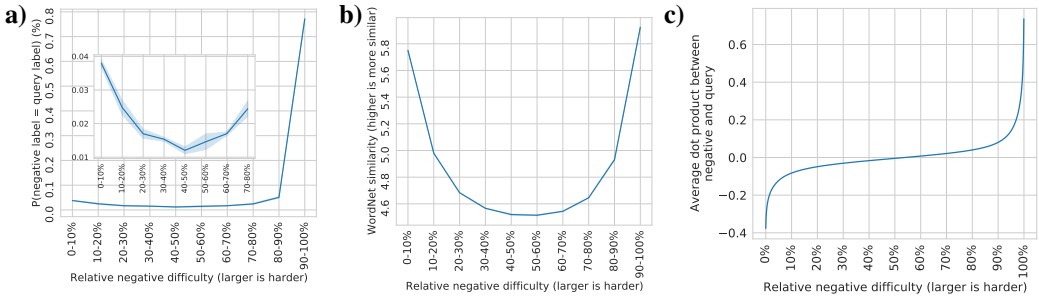

**Figure 5: Semantic similarity with query increases with easier negatives, for the easy half of negatives**
Similarity, as measured by proportion of shared labels (higher is more similar) (**a**) and depth of least common ancestor in WordNet tree (higher is more similar) (**b**) increases with easier negatives, for the easy half of negatives. Similarity also increases with negative difficulty for the harder half of negatives. Average negative distance is negative for the easy half of negatives (**c**). Error bars are standard deviation on plot values across 3 seeds (for b and c, error bars are so small that they are not visible).

consistently of closely related classes such as similar breeds of dog (see Table A4). In contrast to the hard negatives, however, these easiest negatives do not contain many negatives of the same class as the query, although there is a slight increase for the very easiest negatives (see inset, Figure 5a).

**Some negatives are consistently easy or hard across queries.** The hard negatives drive the majority of learning in CID. However, the negatives are ranked independently for each query, so a hard negative for one query may be easy for another. Alternatively, are there negatives that are consistently hard or easy across queries? To test this, we started by measuring the percentage of queries for which each negative was hard, i.e. in the hardest 5%. In Figure 6, we plot the empirical density of the frequency with which each negative is hard; the median is 5% by definition. As a baseline for comparison, we randomized the negatives for each query to approximate the distribution we would expect by chance (orange in Figure 6). The real data distribution (blue) is broader than that expected by chance, so that there are indeed negatives that are more consistently hard and easy than we would expect by random chance. Perhaps maintaining consistently hard negatives in the queue and removing consistently easy ones could improve learning.

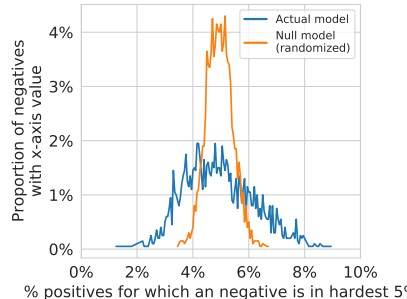

**Figure 6: There exist negatives that are consistently harder or easier than expected by chance.** Distribution of negatives based on the proportion of positives for which a negative is in the hardest 5% for the real data (blue) compared to the distribution obtained by shuffling the negatives for each query (orange).

## 5  RELATED WORK

**Contrastive instance discrimination.** Recently, CID has been utilized in a number of works including NPID (Wu et al., 2018), CMC (Tian et al., 2019), Moco (He et al., 2019), SimCLR (Chen et al., 2020a), MoCo v2 (Chen et al., 2020c), in chronological order. Inspired by its impressive performance, recent works have tried to understand CID from a variety of perspectives. In particular, Tian et al. (2020) investigated the degree of shared information between two augmentations and how it connects to downstream performance, Wang & Isola (2020) suggested that contrastive objectives implicitly try to align similar instances while uniformly utilizing the embedding space, and Arora et al. (2019) proposed a theoretical framework for understanding contrastive learning. Recent work attempted to mitigate the effects of same-class negatives via a reweighting scheme (Chuang et al., 2020), but did not study negatives by difficulty, which is our focus here.

**Non-instance-discrimination self-supervised learning methods.** Beyond CID, a number of other approaches for self-supervised have been proposed that do not work within the CID paradigm, including RotNet (Gidaris et al., 2018), Jigsaw (Noroozi & Favaro, 2016), DeepCluster (Caron et al., 2018), SwAV (Caron et al., 2020), SeLa (Asano et al., 2020), PCL (Li et al., 2020), and BYOL (Grill et al., 2020). Since these did not employ negatives in the same way as CID, our results do not directly relate to these methods.

**Hard negative mining.** There is a recurring theme in the machine learning literature of focusing training on the most difficult examples. In active learning, for example, it is common to favor examples on which the model is most uncertain (Fu et al., 2013). Work in object detection has also benefited from efforts to find hard examples (Sung, 1996; Canévet & Fleuret, 2015; Shrivastava et al., 2016). However, none of the aforementioned work explicitly involved negative examples in the context of CID.

Closest to CID is work on metric learning, where the goal is to learn a representation for each example that is conducive to clustering (Kaya & Bilge, 2019). A standard approach is to use a *triplet loss*, where the loss encourages representing a query (often called an *anchor*) example in a fashion that is close to positive examples from the same class and far from negative examples from other classes (Weinberger & Saul, 2009). In this paradigm, selecting the hardest (Bucher et al., 2016) or harder (Schroff et al., 2015) negatives has improved both the rate of learning and final performance. Similar to our findings about MoCo, Wu et al. (2017) found that mining the very hardest negatives hurts performance (purportedly because it increases the variance of the gradients); they suggested mining harder (but not the hardest) negatives instead.

**Example importance in classification.** In contrast to our work and the aforementioned work on hard negative mining in metric learning, nearly all examples are necessary in image classification. No paper that we are aware of could eliminate more than 20% of examples from CIFAR-10 (Toneva et al., 2018) or 10% from ImageNet (Vodrahalli et al., 2018; Birodkar et al., 2019) without decreases in accuracy. However, not all examples are learned at the same time: the networks learn "easy" examples first (Arpit et al., 2017; Mangalam & Prabhu, 2019) and "hard" examples later in training. However, our notions of easy, hard, and necessary are different than this work: we determine these qualities on a per-query basis (meaning different examples can be easy or hard for different queries) while this work assigns these qualities to specific examples for all of training or across training runs.

## 6 DISCUSSION

Negatives are critical to CID. We studied the relative importance of subsets of negatives, by difficulty, in the context of MoCo v2 Chen et al. (2020c), and found that negative importance varied dramatically by difficulty (Figure 1): the vast majority (easiest 95%) of negatives were insufficient without the hardest 5%, and were unnecessary when those 5% were included (Section 3). Moreover, we found that the very hardest negatives were unhelpful or even harmful to performance and that this could be accounted for by an over-representation of same-class negatives. To understand why the hard negatives are helpful, we showed that the hard negatives are more semantically similar to the query than the easy negatives (Section 4). We also found that there exist easy negatives that are both anti-correlated and semantically similar to the query, and that some of the negatives are consistently easy or hard across queries. Many of these observations are in line with what has been found in other contexts on hard negative mining for metric learning, where accuracy and sample complexity have improved through judicious negative selection methods. Insights from our work may motivate approaches that yield benefits of a similar nature in CID.

### 6.1 LIMITATIONS AND FUTURE WORK

While we focused our experiments on MoCo v2, similar results may be observed for other CID frameworks. However, we leave this to future work along with a study of other downstream tasks. It is also possible that the lessons learned here may be useful for non-CID based contrastive approaches such as SwAV (Caron et al., 2020) and PCL (Li et al., 2020).

One of our most surprising findings was that there exist negatives that are anti-correlated with the query and also more semantically similar to it than average. This seems undesirable for a downstream task of linear classification. Why would the network learn to anti-align two closely related concepts? Understanding the role of such negatives and discovering whether this behavior can be exploited or corrected could be an important direction for future work.

Another avenue for future investigation involves exploring the use of curricula for negative difficulty. For example, a larger quantity of easy negatives may be useful during the early stages of training while harder negatives are more useful later. While developing a negative curriculum is beyond the scope of this work, curricula have shown utility in many other contexts (Bengio et al., 2009).

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

# A   APPENDIX

## A.1   ADDITIONAL NECESSITY/SUFFICIENCY RESULTS

| Train on only | 85-90% | 90-95% | 95-100% |
|---|---|---|---|
| Top-1 accuracy (%) | $19.47 \pm 12.83$ | $51.89 \pm 1.00$ | $66.69 \pm 0.16$ |
| Top-5 accuracy (%) | $36.78 \pm 17.93$ | $75.44 \pm 0.74$ | $87.35 \pm 0.09$ |

| Train on only | 85-100% | 90-100% | 95-100% |
|---|---|---|---|
| Top-1 accuracy (%) | $67.22 \pm 0.21$ | $67.15 \pm 0.10$ | $67.32 \pm 0.88$ |
| Top-5 accuracy (%) | $87.67 \pm 0.09$ | $87.60 \pm 0.02$ | $87.52 \pm 0.63$ |

Table A1: **Extended sufficiency results, 3 seeds each.**

| Train on all except | 85-90% | 90-95% | 95-100% |
|---|---|---|---|
| Top-1 accuracy (%) | $67.56 \pm 0.12$ | $67.53 \pm 0.20$ | $62.1 \pm 0.24$ |
| Top-5 accuracy (%) | $87.98 \pm 0.12$ | $87.94 \pm 0.12$ | $84.0 \pm 0.15$ |

| Train on all except | 85-100% | 90-100% | 95-100 % |
|---|---|---|---|
| Top-1 accuracy (%) | $47.91 \pm 0.79$ | $56.96 \pm 0.36$ | $61.95 \pm 0.16$ |
| Top-5 accuracy (%) | $72.13 \pm 0.83$ | $80.14 \pm 0.20$ | $83.87 \pm 0.28$ |

Table A2: **Extended necessity results, 3 seeds each.**

## A.2   SIMCLR RESULTS

An abridged set of experiments were run on SimCLR Chen et al. (2020a), with one seed each. SimCLR was reimplemented in pytorch and tested with modifications analogous to experiments elsewhere in this paper on MoCo v2, where only a portion of negatives, based on their hardness, were used for pre-training. Pre-training was run for 200 epochs with batch size 4096 on a Resnet-50. For training on the hardest 5% of negatives (95-100% in the table), the first pre-training epoch was trained on all negatives, and all further pre-training epochs were trained on the hardest 5%; pre-training on just the hardest 5% of negatives for all epochs did not converge.

| Train on only | 85-90% | 90-95% | 95-100 % |
|---|---|---|---|
| Top-1 accuracy (%) | 55.60 | 62.70 | 66.34 |
| Top-5 accuracy (%) | 80.46 | 85.30 | 87.43 |

| Train on only | 0-90% | 0-95% | 0-100 % |
|---|---|---|---|
| Top-1 accuracy (%) | 52.43 | 63.66 | 66.99 |
| Top-5 accuracy (%) | 78.46 | 86.06 | 87.73 |

Table A3: **SimCLR results, 1 seed each.**

For MoCo v2, the hardest 5% of negatives had been found to be both necessary and sufficient for downstream linear classification accuracy (Figure 2). We find analogous results for SimCLR: the accuracy for training on just the top 5% of negatives by difficulty (66.34%) is very close to the accuracy for training on all negatives (66.99%), while there is a drop of several percentage points when training on the bottom 95% of negatives by difficulty (63.66%). This suggests the most difficult 5% of negatives are both necessary and sufficient for SimCLR, in addition to MoCo v2.

## A.3 MOST CORRELATED AND MOST ANTI-CORRELATED CLASSES

| Mean dot product | Negative Class | Positive Class |
|---|---|---|
| -0.591357 | Ibizan hound, Ibizan Podenco | keeshond |
| -0.572822 | Italian greyhound | Kerry blue terrier |
| -0.562565 | macaw | ruddy turnstone |
| -0.494559 | Staffordshire bullterrier | affenpinscher |
| -0.487417 | box turtle, box tortoise | nematode |
| -0.476078 | briard | refrigerator |
| -0.471706 | Border collie | Mexican hairless |
| -0.467100 | dalmatian | chow, chow chow |
| -0.460264 | sports car | steam locomotive |
| -0.459015 | Staffordshire bullterrier | Tibetan terrier |

Table A4: **Most anti-correlated classes.** Mean dot product was computed pairwise across each pair of classes.

| Mean dot product | Negative Class | Positive Class |
|---|---|---|
| 0.923779 | monarch | daisy |
| 0.901869 | ground beetle | dung beetle |
| 0.856066 | rifle | rubber eraser |
| 0.823796 | entertainment center | home theater |
| 0.798866 | minibus | police van |
| 0.795254 | bee | monarch, |
| 0.794521 | maillot | swimming trunks |
| 0.789350 | airliner | wing |
| 0.789099 | altar | organ, pipe organ |
| 0.786902 | dogsled | ski |

Table A5: **Most correlated classes.** Mean dot product was computed pairwise across each pair of classes.

| Mean dot product | Negative Class | Positive Class |
|---|---|---|
| -1.112930e-07 | hog | totem pole |
| -2.239249e-07 | canoe | tennis ball |
| 6.617499e-07 | Great Pyrenees | knot |
| 6.956980e-07 | magpie | Cardigan |
| -7.122289e-07 | china cabinet | running shoe |
| -7.863385e-07 | spiny lobster | balance beam |
| 8.588731e-07 | screwdriver | sunglasses |
| -8.760835e-07 | limpkin | packet |
| 8.906354e-07 | impala | coho |
| -9.792857e-07 | boathouse | television |

Table A6: **Most orthogonal classes.** Mean dot product was computed pairwise across each pair of classes.

