# OpenReview forum: "Are all negatives created equal in contrastive instance discrimination?"
_ICLR.cc/2021/Conference — Reject_

### Official Review · AnonReviewer4 · 2020-10-26
**Nice work, but needs more experiments or some theoretical justification**

**Rating:** 5
**Confidence:** 5

**Review:**

In this paper, the authors carried out a series of experiments to analyze the impact of negative samples in contrastive learning (instance discrimination - CID). In particular, they try to identify which difficulty range is important for representation learning. Of the many recent self-supervised learning approaches, they chose MOCO V2 as the testbed. They trained the MOCO model from an ImageNet pre-trained one. Various settings, which correspond to various ways of filtering our hard or easy negatives, were used. Hardness of samples are measured based on embedding distance to the query. I.e. ones with large distance are easy. Their main findings are, for negative samples, 1) Using the 5% hardest is enough for downstream tasks, 2) the easiest 95% of them were unnecessary and insufficient, 3) The hardest 0.1% is harmful and  4) hard negatives were more semantically similar to the query.
In general, in my opinion, this is a paper in which the authors tried to answers many interesting practical questions. The author provided experiments and convincing evidences for a number of insights. My main reservations with this paper are:
1) most of the points are not new and are elaborations of what were pointed out before elsewhere, for example, in semi-hard mining for distance metric learning.
2) the empirical results are only within the context of MOCO2 and for a linear classification task. It is not clear how such numbers as 0.1%, 5% or 95% would change when adopting other frameworks such as BYOL or SwAV… The reported gains seems a little bit sensitive to the temperature parameters of MOCO.
3) The sample hardness is measured based on embedding distance, which would be evolved during the training process itself.  It is not clear how accurate it is especially in the early stage of training.

My suggestion for improvements is that either to empirically show that their findings (numbers) are consistent across a number of frameworks and downstream tasks, or to provide some theoretical justification for their findings if only MOCO v2 is used.

---

> ### Author Response · Authors · 2020-11-25
> **Rebuttal response**
>
> We thank the reviewer for their time and comments. We are glad to hear that you think that our work tries to answer many interesting practical questions, and that our work provides convincing evidence for a number of insights.
>
> We address your comments in order:
>
> 1. We agree that there is a long list of hard-negative mining works in object detection. In fact, we note this explicitly in the paper. What differentiates our work from previous works is that this work does an empirical deep dive in the evaluation of negative importance for recent high-performing CID methods. In doing so, we find the surprising result that only the 5% of hardest negatives drive learning.
>
> 2. We have also performed experiments on SimCLR, another competitive CID method, and found very similar results, where the hardest 5% of negatives were both necessary and sufficient for good linear classification accuracy; the results have been added to the appendix of the paper. We focused our attention on only the highest performing CID methods, since our aim was to understand the role of negatives in the new CID methods with the best performance. Note that SwAV and BYOL are not CID: BYOL is not contrastive, and SwAV is not instance-based. Thus our methods for evaluating negatives do not apply for those methods. This had been explained in the related works section. Additionally, in the introduction, all of the methods had been introduced as SSL methods, but only MoCo and SimCLR as CID. To be extra clear, we have edited the paper to contain another note about this in a footnote in the introduction. Regarding temperature, sensitivity to temperature is in fact one of our findings. Because we seek to understand negatives in the context of recent empirical successes of CID, we focus on regimes in which these CID methods perform well, which is a narrow temperature range.
>
> 3. We agree that difficulty is a function of network weights, and this is by design: we aimed to assess negatives by the difficulty that they were perceived as by the current network. It is significant and interesting that even if the current network is untrained (and unreliable) in early epochs, training on only “hard” negatives according to the current network performs well.

---

### Official Review · AnonReviewer1 · 2020-10-27
**Good empirical analysis, but findings are not sufficiently novel**

**Rating:** 2
**Confidence:** 5

**Review:**


The findings of this work are that for contrastive learning, most of the negatives deemed easily separable are unnecessary, the most important negatives are somewhere in the top 5% closest to the positive sample, and that some of the exceedingly hard examples are detrimental.

-In general, I felt the main findings of this work to be roughly in line with what we already know about contrastive learning. We can easily look at this work's findings with respect to the soft SVM margin, in that only the examples close to the decision boundary should matter (max margin), but some difficult examples  (the aforementioned exceedingly difficult ones) make the data inseparable, so we allow some violation (slack terms).  While I'm not suggesting that slapping a soft SVM here would solve the problem, there is a large body of SVM-based detection/classification literature that precedes the findings of this work.

-Validity of WordNet as a measure of semantic similarity: Section 4 uses WordNet distances to estimate the semantic similarities between classes by finding their shared subtree root. The deeper the subtree, the more semantically similar. While I do not dispute the claim of the hardest negatives being from semantically similar classes. Different parts of the WordNet synset tree have semantic hierarchies of varying levels of coarseness. A 2 hop distance in one subtree could easily be more of a semantic jump than a 3 hop distance in another.

-The exist prior works dealing with the neglected semantic hierarchies in ImageNet by setting up hierarchical classifiers. An example is [1].

-I would further argue that there's some nuance in the correlation between semantic similarity and example hardness, in that it really depends on your choice of feature representation. Visual features will naturally correlate with closer semantic levels in visually-defined categories. However, this will not necessarily hold for semantic categories defined by function, in that two visually distinct items may fall under close semantic labels.

-The related works section claims object detection works have not "explicitly involved negative examples as in CID." I have to imagine this statement is poorly phrased, as [2] (also cited in this paragraph) very explicitly mines for  face-like non-face patterns. There is a very long list of hard-negative mining works in object detection.

Overall, I value the empirical impact of this work, in that the rather detailed analysis may lead to improvements to future versions of the contrastive feature learning task. However, I do not find the findings of this work to be sufficiently novel for this conference, and therefore cannot recommend this work for acceptance in its current state.



[1] Yan et al. HD-CNN: Hierarchical Deep Convolutional Neural Networksfor Large Scale Visual Recognition. ICCV 2015

[2] Sung and Poggio. Example-Based Learning for View-Based Human Face Detection. TPAMI 1998

---

> ### Author Response · Authors · 2020-11-25
> **Rebuttal response**
>
> We thank the reviewer for their time and comments. To respond to each of your points in order:
>
> - In general, I felt the main findings of this work to be roughly in line with what we already know about contrastive learning [...]
>
> We thank the reviewer for their comments. We agree that there are previous works that consider negatives in contrastive learning. However, our work contrasts with prior works in that it empirically evaluates negatives on high-performing recent CID methods, which may differ from methods that have been previously studied, and in particular, are not SVMs; just because something is true for SVMs does not mean that it is true for these CID methods. (Additionally, SVMs are designed to rely on a [hopefully] small number of support vectors, whereas our results finding that CID is driven by a small fraction of negatives is not baked into the objective.)
>
> - Validity of WordNet as a measure of semantic similarity [...] The exist prior works dealing with the neglected semantic hierarchies in ImageNet [...]
>
> We agree that WordNet similarity is not a perfect way to measure semantic similarity. However, just as all models are wrong but some models are useful, we do believe that WordNet similarity allows us to make comparative statements and provides information that is complementary to the most fine-grained, binary similarity metric, of whether two images are in the same class. We also don’t expect this imperfection of WordNet similarity to be biased for the hardest 5% vs the easier 95% of negatives, for example. We have also updated the paper to note this.
>
> - I would further argue that there's some nuance in the correlation between semantic similarity and example hardness [...]
>
> We agree that there is nuance and that example hardness may be a combination of semantic similarity and visual similarities. We also agree this combination may vary by feature representation, and that could be an interesting direction for future research. We have revised the paper to be more clear.
>
> - The related works section claims object detection works have not "explicitly involved negative examples as in CID." [...] There is a very long list of hard-negative mining works in object detection.
>
> We agree that there is a long list of hard-negative mining works in object detection. In fact, we note this explicitly in the paper. What differentiates our work from previous works is that this work empirically evaluates negatives on high-performing recent CID methods, which differ from methods that have been previously studied, and careful empirical evaluation is essential to avoid building on shaky foundations.
>
> - Overall, I value the empirical impact of this work, in that the rather detailed analysis may lead to improvements to future versions of the contrastive feature learning task. However, I do not find the findings of this work to be sufficiently novel for this conference [...]
>
> We believe that although practical advances are important, work on the empirical underpinnings of the phenomena driving CID is also valuable to the field: Empirical rigor is essential, contrastive learning is of increasing consequence and interest, and it's critical to understand the processes that drive learning. By empirically studying the importance of different negatives, we establish a better understanding for future practical improvements. The translation of this knowledge into new methodologies is a logically distinct line of research, which serves as natural next steps. Furthermore, our work serves to provide scientific grounding and further understanding for recently proposed practical advances that do use negatives, for which there is clear interest (see list below).
>
> [1] Anonymous. Conditional negative sampling for contrastive learning of visual representations. Submitted to International Conference on Learning Representations, 2021a. URL https://openreview.net/forum?id=v8b3e5jN66j. under review.
>
> [2] Anonymous. Contrastive learning with hard negative samples. Submitted to International Conference on Learning Representations, 2021b. URL https://openreview.net/forum?id=CR1XOQ0UTh-. under review.
>
> [3] Yannis Kalantidis, Mert Bulent Sariyildiz, Noe Pion, Philippe Weinzaepfel, and Diane Larlus. Hard negative mixing for contrastive learning, 2020.
>
> [4] Shaofeng Zhang, Junchi Yan, and Xiaokang Yang. Self-supervised representation learning via adaptive hard-positive mining, 2021. URL https://openreview.net/forum?id=aLIbnLY9NtH.

---

### Official Review · AnonReviewer3 · 2020-10-27
**Interesting study, but lacking rigor**

**Rating:** 5
**Confidence:** 4

**Review:**


This paper argues that in contrastive self-supervised learning, different negative instances have different importance. This importance is relevant to the ``difficulty" of negative instances. On ImageNet and MoCo2, the authors show that using the most difficult 5% negative instances can achieve similar performance compared with using all negative instances. However, the most difficult 0.1% of negative instances yield bad performance.

I recommend to reject this paper due to the following major concerns: 1) study is performed on a single dataset, which is not convincing; 2) study is performed on a single method, which casts doubts on whether the conclusions hold for other methods; 3) this study does not seem to have practical value.

While this study is interesting, it lacks rigor, in the following aspects.
1. The study is only performed on a single contrastive self-supervised learning method: MoCo2. It is unclear whether the conclusions hold for other contrastive SSL methods, such as BYOL and many others.
2. The study is conducted on a single dataset: ImageNet. It is unclear whether the conclusions hold for other datasets.
3. Another concern is this study does not seem to have practical value. In each iteration during training, finding the hardest examples for a query needs to calculate the inner-product between this query and all other training examples, which is computationally very heavy.
4. In the author's measure of difficulty, the difficulty is a function of network weights. In early stage of the training, the network weights are random, which implies that the calculated difficulty may be meaningless. Can the authors comment on this?

However, the paper does have a few strong points.
1. The paper is well-written. The organization is clear and the paper is easy to follow.
2. The studied problem is interesting and novel.


Other comments.
1. Figure 5a is difficult to interpret. The author may consider to reorganize it.
2. In Figure 3, only three temperature values were considered, which may not be very convincing.

-----------------------------------------------------------------------------------------------------------------------------------
Update: I read the authors' rebuttal. The authors didn't address my concern "The study is conducted on a single dataset: ImageNet. It is unclear whether the conclusions hold for other datasets."  sufficiently. I would like to keep my original rating.

---

> ### Author Response · Authors · 2020-11-25
> **Rebuttal response**
>
> We thank the reviewer for their time and comments. We are glad you think the paper is well-written and that the problem is interesting and novel.
>
> In response to your comments:
>
> 1) study is performed on a single dataset, which is not convincing;
>
> (See #2 in next section)
>
> 2) study is performed on a single method, which casts doubts on whether the conclusions hold for other methods;
>
> (See #1 in next section)
>
> 3) this study does not seem to have practical value.
>
> We respectfully disagree with this statement. Building new knowledge based on rigorous scientific investigation is a vital step in developing new methods. While not the only way to design new approaches, leveraging this knowledge to improve performance is a different endeavor from our aim in this paper.
>
>
> 1. The study is only performed on a single contrastive self-supervised learning method: MoCo2. It is unclear whether the conclusions hold for other contrastive SSL methods, such as BYOL and many others.
>
> Based on your suggestion, we have also performed experiments on SimCLR, another competitive CID method, and found very similar results, where again the hardest 5% of negatives were both necessary and sufficient for good linear classification accuracy. These results have been added to the appendix of the paper.
>
> Regarding other methods, we focused our attention on only the highest performing CID methods, since our aim was to understand the role of negatives in the new CID methods with the best performance. Note that SwAV, SeLa, and BYOL are not CID: BYOL is not contrastive, and SwAV and SeLa are not instance-based. Thus our methods for evaluating negatives do not apply for those methods. This had been explained in the related works section. Additionally, in the introduction, all of the methods had been introduced as SSL methods, but only MoCo and SimCLR as CID. To be extra clear, we have edited the paper to contain another note about this in a footnote in the introduction.
>
> 2. The study is conducted on a single dataset: ImageNet. It is unclear whether the conclusions hold for other datasets.
>
> We agree that more datasets would be even more convincing, but ImageNet is (for now) the gold standard for image classification, and is also much larger and more comprehensive than other datasets that are commonly used, such as ImageNet-100 (a subset of ImageNet), STL, CIFAR, etc.
>
> 3. Another concern is this study does not seem to have practical value. In each iteration during training, finding the hardest examples for a query needs to calculate the inner-product between this query and all other training examples, which is computationally very heavy.
>
> Note that calculating difficulty is not computationally heavy; the inner products corresponding to difficulty are already calculated in order to calculate the loss. Thus, while our experiments do not reduce compute, they also do not meaningfully increase it.
>
> 4. In the author's measure of difficulty, the difficulty is a function of network weights. In early stage of the training, the network weights are random, which implies that the calculated difficulty may be meaningless. Can the authors comment on this?
>
> We agree that difficulty is a function of network weights, and this is by design: we aimed to assess negatives by the difficulty that they were perceived as by the current network. It is significant and interesting that even if the current network is untrained (and unreliable) in early epochs, training on only “hard” negatives according to the current network performs well.
>
>
> 1. Figure 5a is difficult to interpret. The author may consider to reorganize it.
>
> Thank you. We have rephrased the caption to be more clear.
>
> 2. In Figure 3, only three temperature values were considered, which may not be very convincing.
>
> Because we wanted to understand the role of negatives in the empirical successes of recent CID methods, we chose to focus on settings in which these CID methods perform well, which is a narrow temperature range.

---

### Official Review · AnonReviewer2 · 2020-10-29
**The exploration is useful, and would like to see a corresponding framework to be used in the community.**

**Rating:** 5
**Confidence:** 3

**Review:**

This paper mainly studied how the negative samples can affect the model performance in supervised learning CIO works. Through the experiments, this work has a few interesting findings, including the majority of negative samples are not important for the model learning, only a small subset of hard samples determine the model importance. These hard examples are also closely related with positive samples (more semantically similar).  We can see from experiments that it's very important  to fairly treat negative samples in supervised learning tasks. However, there is no frameworks proposed to help improve the learning representation or speed up the training task.  In general, the readers are more interested in the solutions after realizing the importance of negative samples treatment during the experiments.  It would be necessary to include the corresponding solutions by automatically setup these negatives samples in CID related task.

---

> ### Author Response · Authors · 2020-11-25
> **Rebuttal response**
>
> We thank the reviewer for their time and comments. We are pleased to hear that you think the exploration is useful and that the work has interesting findings. We believe that although practical advances are important, work on the empirical underpinnings of the phenomena driving CID is also valuable to the field: Empirical rigor is essential, contrastive learning is of increasing consequence and interest, and it is critical to understand the processes that drive learning. By empirically studying the importance of different negatives, we establish a better understanding for future practical improvements. The translation of this knowledge into new methodologies is a logically distinct line of research, which serves as natural next steps. Our work also provides scientific grounding and further understanding for recently proposed practical advances involving negatives (see list below).
>
> [1] Anonymous. Conditional negative sampling for contrastive learning of visual representations. Submitted to International Conference on Learning Representations, 2021a. URL https://openreview.net/forum?id=v8b3e5jN66j. under review.
>
> [2] Anonymous. Contrastive learning with hard negative samples. Submitted to International Conference on Learning Representations, 2021b. URL https://openreview.net/forum?id=CR1XOQ0UTh-. under review.
>
> [3] Yannis Kalantidis, Mert Bulent Sariyildiz, Noe Pion, Philippe Weinzaepfel, and Diane Larlus. Hard negative mixing for contrastive learning, 2020.
>
> [4] Shaofeng Zhang, Junchi Yan, and Xiaokang Yang. Self-supervised representation learning via adaptive hard-positive mining, 2021. URL https://openreview.net/forum?id=aLIbnLY9NtH.

---

### Decision · Program_Chairs · 2021-01-07
**Final Decision**

**Decision:**

Reject

**Comment:**

This paper empirically studies the impact of different types of negatives used in recent contrastive self-supervised learning methods. Results were initially shown on Mocov2, though after rebuttal simCLR was also added, and several interesting findings were found including that only hardest 5% of the negatives are necessary and sufficient. While the reviewers saw the benefit of rigorously studying this aspect of recent advances in self-supervised learning, a number of issues were raised including: 1) The limited scope of the conclusions, given that only two (after rebuttal) algorithms were used on one datasets, 2) Limited connections drawn to existing works on hard negative mining (which is very common across machine learning including metric learning and object detection), and 3) Limited discussion of some of the methodological issues such as use of measures that are intrinsically tied to the model's weights (hence being less reliable early in the training) and WordNet as a measure for semantic similarity. Though the authors provided lengthy rebuttals, the reviewers still felt some of these issues were not addressed. As a result, I recommend rejection in this cycle, and that the authors bolster some of these aspects for a submission to future venues.

I would like to emphasize that this type of work, which provides rigorous empirical investigation of various phenomena in machine learning, is indeed important and worth doing. Hence, the lack of a new method (e.g. to address the selection of negatives) was not the basis of the decision. While the paper clearly does a thorough job at investigating these issues for a limited scope (e.g. in terms of datasets), a larger contribution is expected for empirical papers such that 1) we can ensure the generality of the conclusions (across methods and datasets), 2) we have a conceptual framework for understanding the empirical results especially with respect to what is already known in adjacent areas (e.g. metric learning and object detection), and 3) we understand some of the methodological choices that were made and why they are sufficiently justified.